# The Use of Biodrying to Prevent Self-Heating of Alternative Fuel

**DOI:** 10.3390/ma12183039

**Published:** 2019-09-19

**Authors:** Teresa Gajewska, Mateusz Malinowski, Maciej Szkoda

**Affiliations:** 1Institute of Rail Vehicles, Faculty of Mechanical Engineering, Cracow University of Technology, al. Jana Pawła II 37, 31-864 Kraków, Poland; maciej.szkoda@mech.pk.edu.pl; 2Department of Bioprocesses Engineering, Energetics and Automatization. Faculty of Production and Power Engineering, University of Agriculture in Cracow, ul. Balicka 116b, 30-149 Kraków, Poland; mateusz.malinowski@urk.edu.pl

**Keywords:** alternative fuels, biodrying, municipal solid waste

## Abstract

Alternative fuels (refuse-derived fuels—RDF) have been a substitute for fossil fuels in cement production for many years. RDF are produced from various materials characterized by high calorific value. Due to the possibility of self-ignition in the pile of stored alternative fuel, treatments are carried out to help protect entrepreneurs against material losses and employees against loss of health or life. The objective of the research was to assess the impact of alternative fuel biodrying on the ability to self-heat this material. Three variants of materials (alternative fuel produced on the basis of mixed municipal solid waste (MSW) and on the basis of bulky waste (mainly varnished wood and textiles) and residues from selective collection waste (mainly plastics and tires) were adopted for the analysis. The novelty of the proposed solution consists in processing the analyzed materials inside the innovative ecological waste apparatus bioreactor (EWA), which results in increased process efficiency and shortening its duration. The passive thermography technique was used to assess the impact of alternative fuel biodrying on the decrease in the self-heating ability of RDF. As a result of the conducted analyses, it was clear that the biodrying process inhibited the self-heating of alternative fuel. The temperature of the stored fuel reached over 60 °C before the biodrying process. However, after the biodrying process, the maximum temperatures in each of the variants were about 30 °C, which indicates a decrease in the activity of microorganisms and the lack of self-ignition risk. The maximum temperatures obtained (>71 °C), the time to reach them (≈4 h), and the duration of the thermophilic phase (≈65 h) are much shorter than in the studies of other authors, where the duration of the thermophilic phase was over 80 h.

## 1. Introduction

Refuse-derived fuels (RDF) are characterized by a high proportion of cellulosic material, i.e., paper, cardboard, and food leftovers. The term alternative fuels refers to waste materials used for co-processing. Such waste typically includes plastics and paper/card from commercial and industrial activities, waste tires, waste oils, biomass waste, waste textiles, residues from dismantling operations, and hazardous industrial waste [1]. As some materials have both useful mineral content and recoverable calorific value, the distinction between alternative fuels and raw materials is not always clear [2]. These impurities at room temperature, even with low oxygen availability, can be oxidized. The exothermic oxidation reaction contributes to the self-heating of the piles, which, in turn, can lead to self-ignition. Fuel produced from mixed municipal waste tends to self-heat to temperatures as low as 75 ± 5 °C [3]. Hogland and Marques [4] and Yasuhara [5] cite numerous cases of autoignition stored in the form of an alternative fuel pile at a temperature of 65 °C.

Refuse-derived fuel (RDF) is the homogenous portion of municipal solid waste (MSW) and preferred as alternative fuel due to their high calorific value low moisture content [6]. Municipal solid waste (MSW) typically has a calorific value of 8–11 MJ·kg^−1^, while the calorific value of refuse-derived fuel (RDF) ranges from 15 to 20 MJ·kg^−1^ [7].

Alternative fuel auto-ignition processes have been very well described in the literature. Yasuhara et al. [8] and Gao and Hirano [9] presented the cases of the explosion of fuels from waste and the results of studies on simulated self-ignition under controlled conditions. The rise in temperature and the possibility of self-ignition are closely related to time refuse-derived fuel (RDF) storage in pile. Initial research in this area was carried out (in 2013–2014) by, among others, Malinowski and Sikora [10], as well as Malinowski and Wolny-Koładka [11].

In Europe, the number of fires in places of flammable waste storage has been increasing in recent years. The result was, among others, adoption of a new landfill directive by the European Union Parliament in 2018. In 2012, 75 fires of landfills and waste dumps were recorded in Poland, by 2017, there were already up to 132. Fires have increased rapidly since 2015, according to the report of the Central Statistical Office [12,13]. The upward trend continues. In 2018, waste fires occurred twice as often as a year ago. Waste landfilling and storage in piles, of course, also affects other elements of the environment and human life, not only as a result of fires [14,15,16].

Factors that affect the temperature inside refuse-derived fuel (RDF) piles and, thus, the possibility of self-ignition are, among others [4]:particle size (degree of material fineness);organic matter content;moisture content;prism size and height; andpressure in the heap.

In addition, the increase in temperature inside the stored alternative fuel, as in the case of biological waste treatment processes, is implied by microorganisms that emit heat during the processing of organic matter (contained in biodegradable fractions) [17,18,19,20,21]. Biodegradable fractions are the sum of 100% organic, 100% paper and cardboard, 50% wood, 50% textiles, 40% multimaterial waste, and 30% fine fraction, i.e., waste with a grain size below 10 mm [22].

The presence of biodegradable fraction from waste is not a favorable phenomenon for refuse-derived fuel (RDF). The presence of these materials increases the moisture content of the fuel, thus reducing its calorific value. An additional, unfavorable aspect is the occurrence of numerous groups of microorganisms that inhabit these wastes and break down the organic matter they contain. The decomposition of organic matter is accompanied by the emission of thermal energy, which causes an increase in temperature in the refuse-derived fuel (RDF) heap and, as a consequence, may contribute to self-ignition.

Malinowski and Sikora [10] stated on the basis of analyzing the morphological composition of alternative fuel (produced in spring–summer 2013) that the percentage of biodegradable fraction in alternative fuel obtained from mixed municipal waste is about 20%. Analyses carried out by Malinowski and Wolny-Koładka [11] showed that in alternative fuel, the share of biodegradable fraction may be in the range of 11% to 29%, while the time of year does not significantly differentiate this content.

Research conducted by Malinowski and Wolny-Koładka [11] showed the presence of many strains of pathogenic microorganisms in alternative fuel, including: Staphylococcus aureus, Escherichia coli, Salmonella Spp., Enterococcus faecalis, and Clostridium perfringens. The presence of pathogenic factors is a risk for developing illnesses. Infection may occur by inhalation or by direct contact with fuel. Bacteria present in stored fuel can cause skin changes and even bacterial myocarditis and meningitis. The employer should provide access to personal protective equipment, while all employees should comply with health and safety regulations. Alternative fuel storage at the production and storage hall is shown in Figure 1.

Due to the possibility of self-ignition in the pile of stored alternative fuel, various treatments are carried out and allow protecting entrepreneurs against material losses and employees against health loss [23]. Warehouse and production halls are equipped with smoke detectors, infrared cameras, fire extinguishing systems, etc. Alternative fuel is mixed with various sorbents in order to cause specific chemical reactions, among others in the direction of deactivation of microorganisms. Processes that can be used for this purpose include:Ozonation. Mechanically, ozone reacts with polysaccharides, proteins, and lipids, transforming them into low molecular weight compounds as a result of cell membrane rupture. If the ozone dose is high enough, mineralization of released cellular compounds may also occur [24]. The effectiveness of ozonation depends on the type of waste, the dose of ozone, and the pH. Ozone treatment is a very effective method of waste hygienization, but expensive because of the need for an ozone generator.Interaction with ultrasound or microwave radiation. Microwave radiation is a type of electromagnetic radiation with a wavelength ranging from 1 m to about 1 mm. The wave spectrum is between IR and ultra-short wave. As in the case of ozonation, these processes destroy the cell membranes of microorganisms and are, unfortunately, cost-intensive [25]. Paradoxically, as a result of the use of ultrasound and microwaves, an organic substrate is released, which can be a source of easily absorbable organic carbon for microorganisms [26,27,28].Change in pH due to the addition of basic (e.g., pile lime) or acidic compounds. Bacteria require neutral conditions, so their fastest growth occurs at an environmental pH of 6.8 to 7.2 [29]. At pH below 6.6, the rate of bacterial growth is rapidly reduced [30]. Increasing the pH may, in turn, lead to an increase in the concentration of ammonia in the reactor and, as a result, to inhibition of the process [22]. The addition of quicklime to waste by several percentage points causes an increase in pH above 10 and complete and permanent deactivation of microorganisms.Biodrying. One of the processes leading to a decrease in the activity of microorganisms in the waste pile that uses the heat they release is biological drying. The process aims to reduce moisture while maintaining high heat of waste combustion. The waste is heated due to the decomposition of an easily biodegradable part of organic matter [15]. Then, fans are started to extract warm and humid air from the pile, e.g., using drainage pipes and bioreactor systems integrated with biofilters. The maximum temperature achieved in this process is 70 °C, which contributes to the destruction of microorganisms and the disappearance of the biological degradation process [31]. Biodrying is usually used in the biological transformation processes of mixed municipal waste and organic waste [16,17,18,32,33,34]. As a result of the process, a stable, manageable product for cement can be produced [17,18,35,36].

The main purpose of the work is to assess the impact of alternative fuel biodrying on the ability to self-heat of this material. The scope of work includes thermographic assessment of the piles of the stored material (refuse-derived fuel) and analysis of the impact of biodrying on the change of physicochemical properties of alternative fuel. The test covers alternative fuels produced from various materials in three variants.

The research problem undertaken in this work is an attempt to answer the question about the possibility of using the biological drying method in the direction of reducing the fuel’s ability to self-heat. A novelty in the subject matter is the use of a new biodrying method consisting in mixing materials inside an innovative bioreactor, which results in increasing the efficiency of the process and shortening its duration.

## 2. Materials and Methods 

### 2.1. Materials

The test materials were obtained from the MIKI Recykling Ltd. (Cracow, Poland) alternative fuel production installation in Krakow (50.032445247N, 20.061035156E). Alternative fuel in this company is produced in several variants depending on the needs of customers, which are cement plants. Three variants were adopted for analysis:Variant A—fuel produced from mixed MSW.Variant B—fuel produced from bulky waste and derived from mechanical plastic sorting (mainly HDPE, LDPE, PP, PS, PET) and paper—unsuitable for the material recycling process due to the level of organic residue contamination.Variant C—fuel produced from car tires and residues from the mechanical sorting process of plastic (mainly HDPE, LDPE, PP, PS, PET) and paper—unsuitable for the material recycling process due to the level of organic residue contamination.

Test samples were taken in accordance with the method recommended by the European Committee for Standardization, 2006, Characterization of Waste—Sampling of Waste Materials—Framework for the Preparation and Application of a Sampling Plan [30]. The alternative fuel collected for testing is stored in the hall in the form of irregular piles, dimensions: base 6 × 10 m and height up to 3 m. The size of a single particle did not exceed 30 mm. The average temperature inside the production and storage hall during the measurements was 24.5 ± 2.0 °C.

### 2.2. Sampling and Laboratory Tests

Test samples were taken in triplicate each time. Samples were taken before and after the biodrying process. The samples were placed in 2000 mL containers and immediately transported to the laboratory for laboratory tests. Total moisture was determined in accordance with CEN-TS 15414-1: 2006 Solid Recovered Fuels [37]. Determination of moisture content was done using the oven dry method, while that of total moisture by a reference method. The tray was weighed with accuracy to 0.1 g and weighed and distributed approximately 300 g of sample. The sample was placed in a laboratory dryer heated to 105 ± 2 °C. After 60 min, the test tray was weighed. The procedure was repeated until the sample mass did not exceed 0.2% of the previous weighing. The ash content was determined in an ELTRA TGA thermogravimeter (ELTRA GmbH, Haan, Germany). The test consisted of measuring the weight loss of the sample at 550 ± 10 °C. About 1 g of a sample was weighed into crucibles, and a programmed analytical cycle was started. After the test, the result was automatically displayed in the program. The end result is the average of three measurements.

Determination of C, H, S, Cl, and N content was done according to PN-EN 15407:2011 Solid Secondary Fuels—Methods for Determining the Content of Carbon (C), Hydrogen (H)m and Nitrogen (N) and PN-EN 15408: 2011 Solid Secondary Fuels—Methods for Determining the Content of Sulfur (S), Chlorine (Cl), Fluorine (F)m and bromine (Br) [38,39].

Determination of carbon, hydrogen, and sulfur content was performed in an ELTRA CHS-580 elemental analyzer (ELTRA GmbH, Haan, Germany). About 100 mg of a sample was weighed into a ceramic boat, which was then placed directly in an oven heated to 1350 °C, where the sample was burned, and the exhaust gas was directed to the measuring cuvettes. The end result is the average of three measurements. The nitrogen content was determined on the ELTRA N-580 analyzer (ELTRA GmbH, Haan, Germany). Approximately 40 mg of sample was weighed into a tin capsule, and the capsule was rolled up and placed in an oven heated to 950 °C. After the analysis, the results of the percentage N were displayed automatically in the program. The end result is the average of three measurements.

Determination of heat of combustion and the calorific value was according to PN-EN 15400:2011 Solid Secondary Fuels—Determination of Calorific Value [40]. The determination was carried out in an IKA POL 5000 calorimeter (IKA POL, Warszawa, Poland). About 1 g of sample was weighed into a crucible and placed in a calorimeter bomb. The bomb was placed in the calorimeter holder, and after introducing the weight, the measurement was started. After the measurement, the heat of combustion value is displayed on the calorimeter panel.

The content of heavy metals in the samples was determined using ICP-OES model 5100 SVDV inductively coupled plasma mass spectrometry system (Agilent Technologies, Santa Clara, CA, USA). Samples for determination of heavy metal content were mineralized in aqua regia.

### 2.3. Biodrying Process

The biological drying process was carried out in the ecological waste apparatus (EWA) bioreactor (Solbien, Ostrava, Czech Republic), which consists of a thermally insulated 36 m^3^ working space, charge aeration system, automatic charge mixing system consisting of a segmented floor, and a tray conveyor located along the inner perimeter of the bioreactor and integrated device for loading and unloading the material subjected to biological drying [41]. The device diagram is shown in Figure 2.

The process of biological drying consists in supplying fresh air to the materials placed in the bioreactor and periodic mixing of the charge, which results in better aeration, thus allowing faster temperature increase and decomposition of complex organic substances contained in the waste. In the first stage of the process, the temperature rises to over 70 °C, which is the result of high metabolic activity [42,43]. High temperatures cause denaturalization of proteins, and after some time, the deactivation of bacteria and pathogenic organisms takes places, which in consequence leads to the sanitization of the dried material and a gradual decrease in the process temperature [31]. The process in the ecological waste apparatus (EWA) bioreactor is controlled based on temperature measurement at four measuring points and oxygen content measurements.

### 2.4. Thermographic Analysis

Thermographs are defined as the recording of infrared radiation (sent by the test object) and converting it in a thermographic camera into a digital image. The camera’s optical system detects thermal radiation, which in the detector is transformed into an electric signal that changes depending on the intensity of infrared radiation. The use of a thermographic camera allows non-contact temperature measurement.

A FLIR ThermaCAM e300 thermographic camera (FLIR Systems, Inc., Portland, OR, USA) was used to measure the temperature of the analyzed RDF fuel. The camera’s temperature resolution is 0.1 °C. Each image has a 16-bit color depth and 320 × 240 pixels, so the resulting data matrices have information about a temperature of 76,800 points. An emissivity factor of 0.95 was adopted for the study, determined in studies carried out by Malinowski and Wolny-Koładka [11]. During the thermographic measurement, the temperature and relative humidity of the air were monitored using a THM-201LP microprocessor thermohygrometer (manufactured by Geneza, Kraków, Poland). Thermograms were made for transverse piles profiles exposed using an excavator. This allowed the visualization of temperatures inside stored piles of alternative fuel before and after the biodrying process. A total of 36 thermograms were analyzed. Analyzes with the thermohygrometer probe was necessary for control of initial temperature. However, the analysis made with a thermographic camera was definitely more precise and allowed for a more accurate determination of the maximum temperature in the pile.

## 3. Results

### 3.1. Characteristics of Raw Materials

Alternative fuel produced from mixed MSW should have appropriate quality standards to ensure environmental protection [45]. The key parameters of alternative fuel that determine its suitability for combustion in cement kilns are its calorific value (>14 MJ·kg^−1^), moisture content (<15%), chlorine content (<0.8%), sulfur (<2.5%) and ash (<15%), polychlorobiphenyl hydrocarbons (PCB) content (<50 mg·kg^−1^), and heavy metal content (<2500 mg·kg^−1^) [36]. Due to the heterogeneous morphological composition of municipal waste (depending on many factors), compliance with the above requirements is difficult [31]. In the case of mixed municipal waste, the separation of the so-called flammable fraction does not predispose it for use in the energy sector [10] and cement plants.

Table 1 and Table 2 summarize the basic energy and chemical characteristics of alternative fuel taken for testing. Alternative fuel intended for incineration in a cement plant is produced in three variants: (A) 100% mixed municipal waste, (B) plastic and paper (50% ± 8%) and bulky waste (50% ± 8%), (C) made of plastic and paper (75% ± 11%) and tires (25% ± 11%). Each of the adopted variants responds to the needs of a specific recipient of alternative fuel (cement plant) [7].

Variants of alternative fuel produced differ mainly in moisture content and heat of combustion. Fuel produced from materials in variant C has the best energy properties (highest calorific value). The calorific value of refuse-derived fuel (RDF) produced from materials in option A is lower than required by cement plants. The fuel produced in variants B and C, despite very favorable energy characteristics, is characterized by too high chlorine content (Table 1). Application as a material component of tires also negatively affects sulfur content.

The analysis of the data contained in Table 2 shows that fuels produced from tires and plastics exceed the permissible content of heavy metals in fuel (variant C).

In each produced waste, the presence of organic inclusions was found, and the heating of the wastes was noted, which was the result of the development of aerobic microorganisms present in this waste. The refuse-derived fuel (RDF) temperature at the time of its generation (after final fragmentation) was on average 26.0 ± 2.2 °C, similarly to the Malinowski and Wolny-Koładka studies [11].

### 3.2. Impact of Biodrying on Refuse-Derived Fuel (RDF) Properties

In all analyzed materials, it should be stated that the biological processing took place and was characterized by a proper course (Figure 3). Thermophilic temperature (i.e., above 45 °C) was observed for all replicates for each type of mixture, which indicates a high microbial activity in the processed materials. The thermophilic phase in all variants began very quickly, i.e., at the latest in the first four hours of the process, which was the result of mixing refuse-derived fuel (RDF) inside the ecological waste apparatus (EWA) reactor. The thermophilic phase usually begins only on day 2 of the process [11].

The maximum process temperature exceeded the value of 71 °C for refuse-derived fuel (RDF) from materials in variants A and B, i.e., alternative fuel produced on the basis of municipal waste and on the basis of large dimensions as well as plastics and paper. In variant C, the maximum process temperature was 63.4 °C, and the thermophilic phase lasted the shortest (49.2 h). In variants A and B, the thermophilic phase lasted, respectively, 65.3 and 64.2 h.

The time to reach maximum temperatures and the duration of the thermophilic phase is much shorter than in the studies of Baran et al. [46] and Tom et al. [42], where the duration of the thermophilic phase was over 80 h, with the total duration of the bios drying process over seven days.

Despite the fact that the process lasted only four days and the thermophilic phase was shorter than in similar studies, as a result of the refuse-derived fuel (RDF) biodrying process in variants A and B, the moisture content of the materials decreased by 53.1% and 49.3%, respectively. Moisture content for variant C after the biodrying process is higher due to the conditions of sample collection. The ash and carbon content slightly decreased (statistically insignificant difference). These values have not changed due to the short duration of the process. Fuel produced from materials in variants A and B significantly increased its energy value (Table 3). The calorific value of the end product over 18 MJ·kg^−1^ was observed to be similar to those in [17,18,47]. The heat of combustion and calorific value for variant C remained at the same level after the biodrying process (the difference was not statistically significant). Moreover, the calorific value obtained for variant A after the biodrying process guarantees the sale of alternative fuel to the cement plant. Alternative fuel produced in variant A should be landfilling at the landfills, which generated cost. The company’s annual savings could be PLN 200/Mg (around €50).

Figure 4a shows an example of a thermogram made for alternative fuel (variant B) that has not undergone the biodrying process. The thermogram was made after 48 h of storage in a pile. The thermograms made were the basis for determining the maximum temperatures inside the stack of stored material at various times from the moment they were manufactured. It is important that the temperature inside the pile not be uniform. The highest temperatures occurred in the upper zone from 0.5 to 1.0 m below the surface of piles.

Figure 4b shows the thermogram of refuse-derived fuel (RDF) subjected to biodrying in the ecological waste apparatus (EWA) bioreactor. The thermogram was made after 48 h of storage in the pile from the moment of discharge from the bioreactor. Table 4 presents a comparison of temperatures achieved in the alternative fuel heap before and after the biological drying process. The maximum observed the temperature of the fuel self-heating process in the pile was 68.2 °C and for the analyzed duration of the process was slightly lower compared to the results of the tests of Hogland and Marques [4] and Malinowski and Wolny-Koładka [11] (over 75 °C). The refuse-derived fuel (RDF) self-heating process, until the temperature reached 65 °C, occurred slower in the pile than in the case of sewage sludge [48] (about 24 h) and much slower than in the case of mixed municipal waste (over 40 h) [17,18].

As a result of the analysis of thermograms, it was clear that the process of biological drying inhibited the process of fuel self-heating. On this basis, it was determined that the material produced is safe and can be stored on site before transport to the final development facility. Before the biodrying process, the temperature of the stored fuel reached a value of over 60 °C, which may cause self-ignition. After the biodrying process, the maximum temperatures in each of the variants were about 30 °C, which indicates low activity of microorganisms. Malinowski and Wolny-Koładka [31] state that biodrying is an effective method of material hygienization, which was confirmed by the analysis of thermograms made.

## 4. Discussion

Usually, the time span necessary to dry refuse-derived fuel (RDF) is seven days [49], while in industrial conditions, it is 7–15 days [47]. Industrial research in the field of biodrying is directed mainly to the search for methods of biodrying which will allow reducing its duration. As presented in the article, this is possible as a result of the innovative EWA technology. As a result of using the refuse-derived fuel (RDF) biodrying process in the EWA bioreactor after 80 h of the process, stable fuel without any heating characteristics was obtained. In addition, extra benefits were achieved, particularly in terms of the increase in calorific value of refuse-derived fuel (RDF) produced in variants A and B. Therefore, the use of the EWA reactor is also of economic importance. Higher calorific value of the material translates directly into greater financial profits. In addition, compared to other methods of preventing self-heating of RDF, one does not need to buy CaO, generate ozone or generate ultrasound or microwave waves, which can have a negative effect on the life and health of employees.

As a result of the process, the water content of the materials was reduced by about 50% (variants A and B). In research conducted by Malinowski and Wolny-Koładka [11], Colomer-Mendoza et al. [50], Ma et al. [51], Tom et al. [42], and Mohammed et al. [52], a similar moisture loss was achieved. The high water loss proves the high efficiency of the process achieved in a very short time (four days) in relation to other experiments.

It may also be important that these tests were carried out on a real object. Under laboratory conditions, Velis et al. [15] mention that in the process of biological drying of municipal solid waste for the period of 7–15 days, the moisture content should decrease by at least 20%. Bilgin and Tulun [33] conclude that during a 13-day period of drying mixed municipal solid waste, it is possible to decrease the water content by over 30% by increasing the calorific value. In other works, the semi-industrial rotary drum bioreactor achieved a moisture reduction from 40% to 20% in less than seven days of operation [49]. The commercial process cycles were completed within 7–15 days, with mostly loses of water 25–30% w/w, leading to moisture contents of less than 20% w/w [47].

## 5. Conclusions

The research presented in the work was focused on the search for solutions to prevent the self-heating of alternative fuel produced from various materials. Three refuse-derived fuel (RDF) variants were adopted for the analysis. The study attempted to examine the possibility of using the biodrying method and the innovative ecological waste apparatus (EWA) bioreactor in the direction of decreasing fuel self-heating capacity. The method used in research involving the mixing of materials inside an innovative bioreactor resulted in increased process efficiency and shortened its duration.

As a result of the analyses, it was clear that the biodrying process carried out in ecological waste apparatus (EWA) reactors inhibited the process of fuel self-heating, despite the fact that this process lasted only four days (two times shorter than in other studies), and the duration of the thermophilic phase lasted at least 15 h less than in similar studies. Before the biodrying process, the temperature of the stored fuel reached a value of over 60 °C, which may cause self-ignition. However, after the biodrying process, the maximum temperatures in each of the variants were about 30 °C, which indicates a low activity of microorganisms. Such refuse-derived fuel (RDF) can be safely stored in the hall.

As a result of the biodrying process in variants A and B, the moisture content of materials decreased by 53.1% and 49.3%, respectively. The high water loss proves the high efficiency of the process achieved in a very short time (four days) in relation to other experiments.

In the future, the authors plan to continue research on the use of the biodrying process and supplementation with other materials in the direction of reducing the ability of refuse-derived fuel (RDF) to ignite, especially taking into account the use of different air flows in the reactor. Important from a cognitive point of view is also the need to analyze the process of heating up the fuel in the entire volume of the pile, including in the direction of learning the mechanism of temperature increase.

## Figures and Tables

**Figure 1 materials-12-03039-f001:**
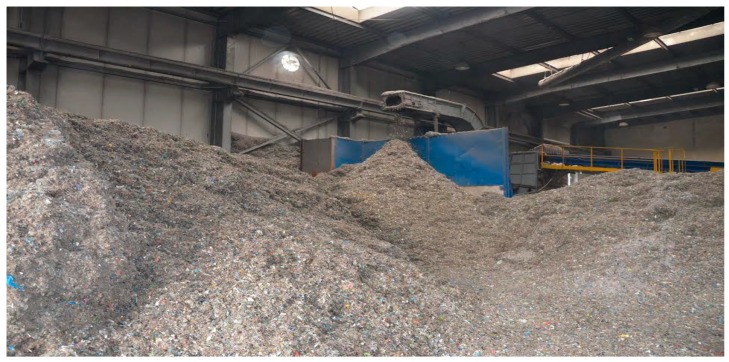
Alternative fuel storage method in the production and storage hall (own photo).

**Figure 2 materials-12-03039-f002:**
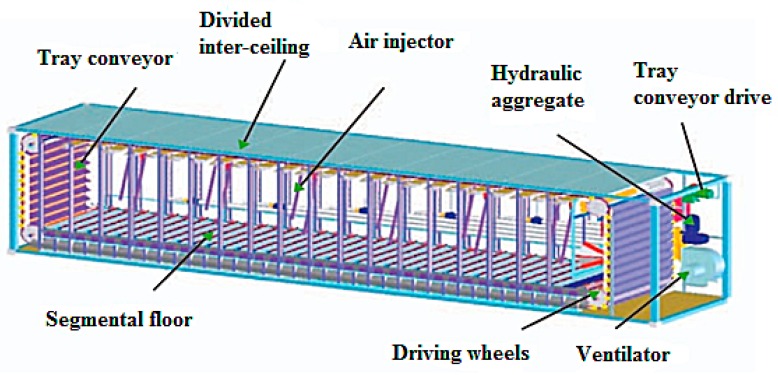
Ecological waste apparatus (EWA) bioreactor cross-section [44].

**Figure 3 materials-12-03039-f003:**
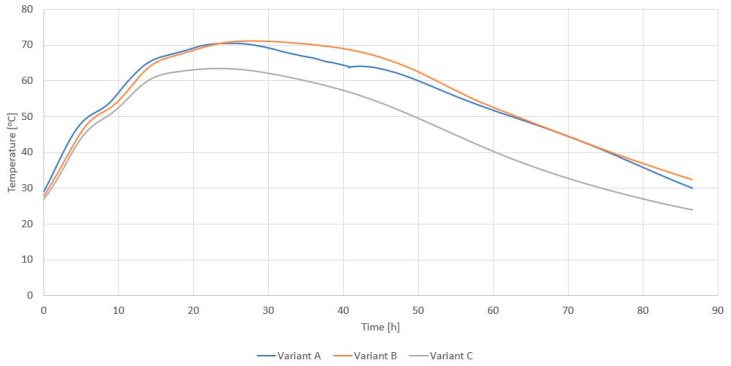
Averaged result of measurements of temperature changes inside EWA bioreactors during refuse-derived fuel (RDF) biodrying with characteristics A, B, and C (own research).

**Figure 4 materials-12-03039-f004:**
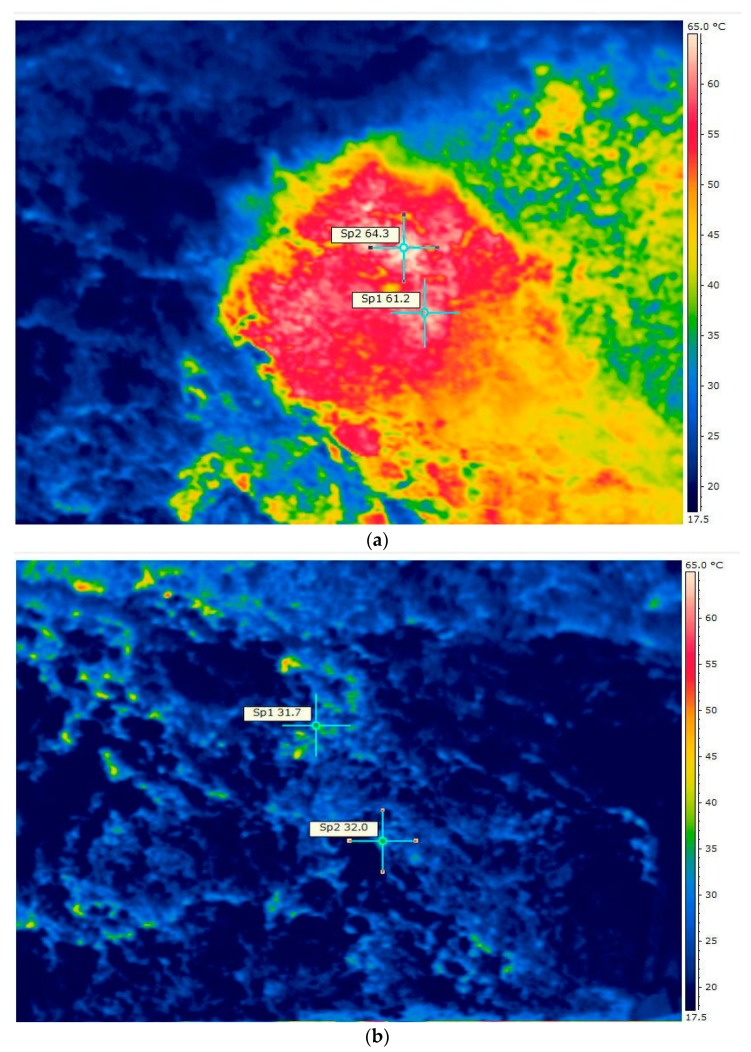
Exemplary thermogram of alternative fuel not subjected to the biodrying process (cross-section) (**a**), and subjected to biodrying (**b**).

**Table 1 materials-12-03039-t001:** List of basic energy characteristics of alternative fuels (own research).

Parameter	Unit	Variants		
-	-	from Mixed Municipal Waste (A)	from Residues from Selective Collection Waste and Bulky Waste (B)	Plastic and Tires (C)
Moisture content	% w/w	25.4 ± 3.2	13.4 ± 0.3	3.1 ± 0.3
Ash content	% d.m.	23.1 ± 0.9	22.6 ± 0.9	11.5 ± 0.7
Sulphur content	% d.m.	0.39 ± 0.08	0.45 ± 0.05	1.66 ± 0.11
Total carbon content	% d.m.	48.3 ± 3.0	53.6 ± 3.1	58.7 ± 4.2
Hydrogen content	% d.m.	6.56 ± 0.71	5.78 ± 0.82	8.76 ± 1.01
Nitrogen content	% d.m.	0.78 ± 0.08	0.70 ± 0.08	0.68 ± 0.08
Heat of combustion	kJ·kg^−1^ d.m.	19,979 ± 1019	23,366 ± 772	31266 ± 819
Calorific value	kJ·kg^−1^	13,833 ± 883	18,762 ± 704	30975 ± 725
Chlorine content	% d.m.	0.65 ± 0.13	1.12 ± 0.22	1.43 ± 0.17

**Table 2 materials-12-03039-t002:** Chemical characterization of alternative fuels (own research).

Element	Unit	A	B	C
As	mg·kg^−1^ d.m.	14 ± 2	11 ± 2	38 ± 4
Ba	mg·kg^−1^ d.m.	272 ± 11	258 ± 9	401 ± 6
Cd	mg·kg^−1^ d.m.	0.8 ± 0.2	3.1 ± 0.3	6.1 ± 0.2
Co	mg·kg^−1^ d.m.	12 ± 1	7 ± 1	11 ± 2
Cu	mg·kg^−1^ d.m.	114 ± 11	592 ± 32	710 ± 31
Cr	mg·kg^−1^ d.m.	136 ± 9	260 ± 12	600 ± 18
Hg	mg·kg^−1^ d.m.	0.9 ± 0.1	0.4 ± 0.1	0.8 ± 0.1
Mo	mg·kg^−1^ d.m.	20 ± 6	59 ± 7	171 ± 8
Ni	mg·kg^−1^ d.m.	9 ± 3	140 ± 3	315 ± 6
Pb	mg·kg^−1^ d.m.	3 ± 2	123 ± 7	25 ± 5
Sb	mg·kg^−1^ d.m.	32 ± 2	32 ± 2	12 ± 2
Se	mg·kg^−1^ d.m.	18 ± 3	39 ± 6	22 ± 6
Sn	mg·kg^−1^ d.m.	14 ± 4	45 ± 3	31 ± 3
Sr	mg·kg^−1^ d.m.	107 ± 4	177 ± 11	412 ± 17
V	mg·kg^−1^ d.m.	12 ± 1	12 ± 1	32 ± 1
Zn	mg·kg^−1^ d.m.	540 ± 17	600 ± 12	633 ± 10

**Table 3 materials-12-03039-t003:** Summary of the basic energy characteristics of refuse-derived fuel (RDF) after the biodrying process (own research).

Parameter	Unit	Variants of Alternative Fuel Produced		
-	-	from Mixed Municipal Waste (A)	from Residues from Selective Collection Waste and Bulky Waste (B)	Plastic and Tires (C)
Moisture content	% w/w	11.9 ± 1.0	6.8 ± 0.8	4.7 ± 0.8
Ashes content	% d.m.	22.5 ± 0.2	21.3 ± 0.3	12.6 ± 0.2
Sulphur content	% d.m.	0.23 ± 0.04	0.70 ± 0.12	1.59 ± 0.05
Total carbon content	% d.m.	48.5 ± 1.5	53.7 ± 1.1	55.1 ± 1.8
Hydrogen content	% d.m.	6.1 ± 0.2	7.6 ± 0.3	8.8 ± 0.2
Nitrogen content	% d.m.	0.65 ± 0.17	0.47 ± 0.16	0.63 ± 0.21
Heat of combustion	kJ·kg^−1^ d.m.	20,848 ± 156	23,540 ± 334	30962 ± 152
Calorific value	kJ·kg^−1^	18,439 ± 155	20,686 ± 301	30556 ± 140
Chloride content	% d.m.	0.80 ± 0.11	1.14 ± 0.09	1.50 ± 0.08

**Table 4 materials-12-03039-t004:** Distribution of maximum temperatures achieved in the alternative fuel piles before and after the biological drying process (°C).

Variant		Maximum RDF Temperature [°C] during Storage on the Pile after:				
		6 h	12 h	24 h	48 h	72 h
A	Before biodrying	26.8	30.6	45.4	66.5	68.2
	After biodrying	30.2	29.7	31.2	30.9	29.6
B	Before biodrying	25.4	29.9	42.8	62.9	66.2
	After biodrying	30.0	29.8	28.7	28.6	27.2
C	Before biodrying	24.6	29.4	39.7	48.8	52.6
	After biodrying	24.6	25.7	24.3	25.2	24.9

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
