# Peer review of "The Use of Biodrying to Prevent Self-Heating of Alternative Fuel"

_materials, 2019, doi:10.3390/ma12183039_

Round 1
Reviewer 1 Report
The main goal of this research study was to determine the drying effect on self-heating of RDF fuels. Conception and processing of this paper is on a suitable level with acceptable scientific background, and in general I can evaluate this paper positively. However I can say that it is not presented the efficiency of lonely drying process (expensive treatment) comparing the gains of this research study.
My recommendations for authors:
1.) Authors should change or to add some more arguments. It mainly concerns the final evaluation regarding extra benefits, where authors describes, that this process have greater financial profits. Where is the prove for this claim? Did authors some economical calculations?
2.) Units should be added to the Table 4.
3.) The process of temperature (in pile) measuring should be more described and I guess (for the future) it should be changed. My opinion is that it is more precisly and fairer to use some contact measuring method instead of thermomeasuring. With some rod sensor they are able to determine temperatures in different places in a pile.
4.) For future research I recommend to extend the storage duration on pile. Maybe the temperature in pile (after drying) will increase later.
Author Response
Dear Reviewer,
We would like to thank the reviewer for his/her kind comments. At the same time, we send the answers to the comments contained in the review form in the attachment.

Reviewer 2 Report
The manuscript entitled „The use of biodrying to prevent self-heating of alternative fuel”, has in attention the prevention of self-ignition of the alternative fuels by using a bio-drying process.
The manuscript needs a careful check of the English language and a thorough spell-check. Moreover, the manuscript needs to be rewritten for a better understanding of the research objectives and of the results.
It is commendable to replace Figure 1 with a figure that is relevant for the bio-drying process.
The authors are kindly requested to:
mention the meaning of the abbreviation where it first appears;
correct the degree sign everywhere in the manuscript;
renumber the references in the manuscript (in the present form the numbering in the text it starts at 15);
provide the standard deviations in Table 2;
include in Figure 4 the results for sample A, B, and C without bio-drying;
Please comment on the results presented in table 2 and 4, before and after bio-drying. It seems that in the case of sample C the moisture content is increasing after the bio-drying.
The manuscript should also include discussions/conclusions regarding the results for sample C.
Author Response

(The authors gave the same response as above.)

Reviewer 3 Report
The submitted study is aimed at assessment of the impact of alternative fuel bio-drying on the ability to self-heat the material. For this purpose, the alternative fuel produced from various materials was tested by processing inside the Ecological Waste Apparatus bioreactor.
The studied results proved to achieve the increase process efficiency and shortening its duration. The results are clearly presented by the tables and graphics. But I have some suggestions for the corrections as follows:
The last sentence of the abstract: The specific values of the experimentally determined time to reach the maximum temperature, and the duration of the thermophilic phase by use of the EWA reactor could be mentioned.
The EWA abbreviation should be explained in the abstract.
All the used abbreviations should be defined in parentheses the first time they appear in the main text. Please revision throughout the text. For example the abbreviation RDF, MSW, IR, etc.
Please improve quality of Figure 5. The values on the temperature scale are difficult to read.
Tables 1, 3 extend beyond the right edge of the page.
The English language revision should be done.
Author Response

(The authors gave the same response as above.)

Round 2
Reviewer 2 Report
The authors addressed most of the recommendations, but the references are not properly inserted in the text. For example, reference 71 does not exist, while reference 9 in the text is actually 14 in the bibliography. Same for references 5, 8, 10, 11, 31, etc.
The paper can be accepted after the references are correctly cited.
Author Response
Dear Reviwer,
Thank you for your comments on the literature.
I have changed from 71 to 17. Now it is correct.
All other literature items are correct. Their change results from the fact that according to the commentary of reviewer 1 I changed from alphabetical numbering to the one with the order of quoting in the text. Hence these changes.
Reviewer 3 Report
The authors accepted my comments and suggestions and improved quality of the paper.
Author Response
Dear Reviewer,
Thank you very much.